# Laryngeal *Helicobacter pylori* Infection and Laryngeal Cancer-Case Series and a Systematic Review

**DOI:** 10.3390/microorganisms9061129

**Published:** 2021-05-23

**Authors:** Li-Jen Hsin, Hai-Hua Chuang, Mu-Yun Lin, Tuan-Jen Fang, Hsueh-Yu Li, Chun-Ta Liao, Chung-Jan Kang, Tse-Ching Chen, Chung-Guei Huang, Tzu-Chen Yen, Li-Ang Lee

**Affiliations:** 1Department of Otorhinolaryngology—Head and Neck Surgery, Chang Gung Memorial Hospital, Linkou Main Branch, Taoyuan City 33305, Taiwan; lijen.hsin@gmail.com (L.-J.H.); fang3109@cgmh.org.tw (T.-J.F.); hyli38@cgmh.org.tw (H.-Y.L.); liaoct@cgmh.org.tw (C.-T.L.); handneck@gmail.com (C.-J.K.); 2Faculty of Medicine, Chang Gung University, Taoyuan City 33302, Taiwan; chhaihua@gmail.com (H.-H.C.); demorgan9864@gmail.com (M.-Y.L.); ctc323@cgmh.org.tw (T.-C.C.); yentc1110@gmail.com (T.-C.Y.); 3Department of Family Medicine, Chang Gung Memorial Hospital, Linkou Main Branch, Taoyuan 33305, Taiwan; 4Department of Pathology, Chang Gung Memorial Hospital, Linkou Main Branch, Taoyuan City 33305, Taiwan; 5Department of Laboratory Medicine, Chang Gung Memorial Hospital, Linkou Main Branch, Taoyuan City 33305, Taiwan; joyce@cgmh.org.tw; 6Research Center for Emerging Viral Infections, Graduate Institute of Biomedical Sciences, Chang Gung University, Taoyuan City 33302, Taiwan; 7Department of Nuclear Medicine and Molecular Imaging Center, Chang Gung Memorial Hospital, Linkou Main Branch, Taoyuan City 33305, Taiwan

**Keywords:** *Helicobacter pylori*, laryngeal cancer, laryngopharyngeal reflux, *CD1d*, *E-cadherin*

## Abstract

*Helicobacter pylori* (*H. pylori*) infection involves the development of gastric cancer and may be associated with laryngeal cancer. However, laryngeal *H. pylori* infection in Taiwanese patients with newly diagnosed laryngeal cancer has not been reported. This study was aimed to investigate the possible association between laryngeal *H. pylori* infection and laryngeal cancer in Taiwan and perform a systematic review of previous reports in other countries. An analysis of 105 patients with laryngeal lesions found the positive rates of *H. pylori* DNA (determined by polymerase chain reaction) and antigen (determined by immunohistochemistry) of the laryngeal lesions were relatively low (vocal polyps: 3% and 3%; vocal fold leukoplakia: 0% and 0%; laryngeal cancers: 0% and 2%). Furthermore, *H. pylori*-associated laryngopharyngeal reflux and the expression of *E-cadherin* and *CD1d* (determined by immunohistochemistry) were comparable among the three subgroups. Fifteen studies were involved in the systematic review of the digital literature database, distributed to February 2021. The data of patients with laryngeal cancer and controls showed that the laryngeal *H. pylori* infection rates were 29.4% and 16.7%, respectively. Although current evidence supported that laryngeal *H. pylori* infection was associated with laryngeal cancer globally, it might not play a role in the development of laryngeal cancer in Taiwan.

## 1. Introduction

Laryngeal cancer is one of the most common respiratory cancers worldwide [1]. In 2018, the crude incidences of laryngeal cancer were estimated to be 2.32 per 100,000 persons worldwide [2] and 3.11 per 100,000 persons in Taiwan [3], respectively. The major risk factors for laryngeal cancer include cigarette smoking, alcohol consumption, male sex, and age >55 years old [4].

*Helicobacter pylori (H. pylori)* is a microaerophilic Gram-negative bacterium that lives in the stomach and duodenum. The helical shape, number of flagella, motility, and urease secretion facilitate *H. pylori* to survive in the acidic environment of the stomach [5]. Despite medical advances, the prevalence of *H. pylori* is high worldwide (33–50%) [6]. *H. pylori* is known to play a role in the carcinogenesis of gastric cancer and mucosa-associated lymphoid tissue lymphoma [7]. It is a major risk factor for gastric cancer among East Asian populations [8]. Furthermore, *H. pylori* and Epstein–Barr virus co-infection might be implicated in the development of gastric cancer [9,10]. Host genetic factors, alcohol, capsaicin consumption, ingestion of inflammatory foods, and stress levels are the known risk factors for gastric cancer [11].

Recently, approximately 50% of patients with laryngeal cancer have gastric *H. pylori* infection (prevalence, 47–75%) [12,13,14]; however, some studies have suggested that gastric *H. pylori* infection may not play a role in the development of laryngeal cancer compared with controls [15,16]. *H. pylori* infection can induce laryngopharyngeal reflux (LPR) [17,18,19]; furthermore, gastric reflux can bring it to the larynx. Therefore, *H. pylori* is present in laryngeal mucosa [20] and has the potential to damage epithelial and mucosal barriers, and that the subsequent inflammatory process can lead to chronic harm and epithelial cell proliferation resulting in laryngeal pathology [21]. Although the relationship between *H. pylori* and gastroesophageal reflux disease is not conclusive [22], the presence of GERD is associated with an increased risk of developing laryngeal cancer [23].

Increasing evidence supports that *H. pylori* may play a role in the development of laryngeal cancer, including that (1) *H. pylori* can be detected in the normal larynx [16], vocal polyps (VP) [24], vocal fold leukoplakia (VFL) [25], and precancerous lesions [26], and (2) *H. pylori* can induce systemic inflammation and promote tumorigenesis [26]. A previous systematic review suggested that *H. pylori* infection of the stomach, larynx, or non-specific sites was related to laryngeal cancer [17].

Noteworthily, *H. pylori* infection has been diagnosed by different techniques in those studies [16,17,24,25,26]. Bacterial culture is the most specific way to confirm viable *H. pylori* of the mucosa tissue; however, this method is tedious, time-consuming, and unnecessary for the routine diagnosis of *H. pylori* infection. Other invasive diagnostic methods (such as the rapid urease test [RUT], histological examination, immunohistochemistry [IHC], or polymerase chain reaction [PCR] assay) and non-invasive diagnostic methods (such as the urea breath test, stool antigen detection, or serology) have been widely used to diagnose *H. pylori* infection [27]. However, these methods present non-optimum specificity (e.g., the RUT, urea breath test, and serology) or difficulty in assessing the existence of active infection (e.g., histological examination, IHC, and PCR) [28].

From a molecular genetic perspective, both *H. pylori* infection [29] and LPR [30] are associated with a decreased expression of *E-cadherin*. This can then reduce the apoptosis of HEp-2 cells, which are considered to originate from human laryngeal carcinoma [31]. Furthermore, both *H. pylori* [32] and LPR [33] have been shown to induce an over-expression of *CD1d* in mucosal tissue, and both CD1d and lipid antigens can stimulate CD1d-restricted T cells (i.e., natural killer T cells) to further activate innate and adaptive immune cells in the tumor microenvironment [34]. Therefore, it would be interesting to investigate the difference in *H. pylori*- and/or LPR-associated expression of *E-cadherin* and *CD1d* between various laryngeal lesions.

The role of gastric *H. pylori* infection in the pathogenesis of gastric cancer has been substantially studied. Some research has also suggested an association between laryngeal *H. pylori* infection and laryngeal cancer (inflamed larynx lesions); however, the connection remains controversial [35]. We hypothesized that laryngeal *H. pylori* infection was involved in the development of laryngeal cancer. The aims of this study were (1) to investigate the prevalence of laryngeal *H.*
*pylori* infection in a prospective case series of patients with newly diagnosed laryngeal lesions including VP, precancerous VFL, and laryngeal cancer in our hospital, (2) to investigate the expression of *E-cadherin* and *CD1d* of the larynx in this population, and (3) to elucidate the impact of laryngeal *H. pylori* infection on the development of laryngeal cancer across countries by performing a systematic review on the topic.

## 2. Materials and Methods

### 2.1. Patients

A total of 105 consecutive patients with newly diagnosed, histologically confirmed laryngeal lesions who underwent laryngeal surgery at the Department of Otorhinolaryngology, Head and Neck Surgery, Chang Gung Memorial Hospital, Linkou Main Branch, Taoyuan, Taiwan, between 1 August 2012 and 31 December 2015, were recruited. This study protocol has been reported elsewhere [36,37]. The sample size (*n* = 105) was estimated using a *priori* calculations [Fisher’s exact test, OR = 2.9, two-tailed α = 0.05, power = 0.90; allocation ratio = 0.4], as reported in a previous meta-analysis of *H. pylori* infection in patients with laryngeal cancer and controls [17]. The inclusion criteria were as follows: age >18 years, a pathological diagnosis (VP, VFL, or laryngeal cancer) of a laryngeal lesion, and a willingness to participate in this study. The exclusion criteria were an unwillingness to undergo tissue examinations and/or answer subjective questionnaires. This prospective case series was approved by the Institutional Review Board (number: 100-4421B) at Chang Gung Medical Foundation, Taoyuan, Taiwan. Written informed consent was obtained from all the participants before enrolment. This study was conducted in accordance with the World Medical Association Declaration of Helsinki.

### 2.2. Collection of the Data and Specimens

Age at diagnosis, sex, cigarette smoking status, alcohol consumption status, and LPR symptom score were recorded at enrollment. The LPR symptom score was evaluated with the Reflux Symptom Index (RSI) questionnaire, which is a highly reliable test (correlation coefficient = 0.81) [38] and has been validated to evaluate the significance of LPR in benign laryngeal lesions [39] and early laryngeal cancer [40]. The severities of nine problems over the past month on a scale of 0 (no problem) to 5 (severe problem) with a maximum total score of 45 were rated by the patients. The specimens were collected during the laryngeal surgery. Pathological diagnosis and pathological staging (according to the 2009 revision of the American Joint Committee on Cancer tumor-node-metastasis staging system) [41] were also collected.

### 2.3. Detection of H. pylori DNA in the Laryngeal Lesions

Using hematoxylin and eosin staining, the lesion type and tissue adequacy (≥10.0% mucosal lesion cells) and resected the corresponding formalin-fixed, paraffin-embedded (FFPE) tissues were histologically evaluated [36]. DNA was extracted from three 5-μm-thick FFPE sections per specimen according to the phenol-chloroform extraction method. A PCR assay was used to detect the *ureC* (*glmM*) gene of the *H. pylori* genome [42]. A final volume of 25 mL of PCR mixture containing 1 µL of each primer, 8.5 µL H_2_O, 12.5 µL d iTaq Universal Probes Supermix (BioRad Laboratories, Hercules, CA), and 2 µL of DNA template was used for DNA amplification. Two primers [ureC (136 bp): 5′-AAGCTTTTAGGGGTGTTAGGGGTTT-3′ and 5′-CGCAATGCTTCAATTCTAAATCTTG-3′] were used for amplifying the *ureC* gene. The reaction was carried out in a T-Gradient thermocycler (Biometra, Göttingen, Germany). The detailed PCR protocol has been described elsewhere [42]. After 40 PCR cycles, the PCR products were resolved by 2% agarose electrophoresis and visualized after staining with ethidium bromide (0.5 mg/mL). In the case of the presence of *ureC*, DNA sequencing was performed, and the nucleotides of DNA sequences were compared by using the Basic Local Alignment Search Tool (http://https://blast.ncbi.nlm.nih.gov/ (accessed on 10 May 2021)).

### 2.4. Detection of H. pylori Antigen in the Laryngeal Lesions

All FFPE tissues which had been histologically evaluated for lesion type and lesion tissue adequacy were used for tissue microarray construction as previously described [36]. IHC was used to detect the presence of *H. pylori* antigen (CAT # RBK012; Zytomed Systems, Germany) [43]. Sections of tissue microarrays were stained using an automated immunostainer (BOND-MAX; Leica Biosystems Ltd., Newcastle, UK), according to the manufacturer’s instructions. Tissues of a known *H. pylori*-positive gastric adenocarcinoma were used as a positive control (Figure 1a), while tissues of a known *H. pylori*-negative gastric adenocarcinoma were used as a negative control (Figure 1b). For optimal detection, the antibody was diluted at 1:800. Antibody reactions were performed at room temperature for 10 min.

### 2.5. Measurements of the Expressions of E-cadherin and CD1d in the Laryngeal Lesions

As mentioned above, IHC was used to measure the expressions of *E-cadherin* (CAT # E-CAD-L-CE; Leica Biosystems Ltd., Newcastle, UK) and *CD1d* (CAT # orb11652; Biorbyt, Cambridge, UK). Sections of tissue microarrays were automatically stained. For optimal detection, the antibodies were diluted at 1:100. Antibody reactions were performed at room temperature for 30 min (*E-cadherin*) and 20 min (*CD1d*) [37]. Using an Aperio ScanScope scanner (Leica Biosystems, Richmond, IL, USA), the stained slides were digitized at 40× magnification. The presence of *H. pylori* and the cellular expression levels of *E-cadherin* and *CD1d* were digitally assessed in four specified regions of interest via semi-quantitative image analysis (Tissue Studio v2.1; Definiens AG, Münich, Germany) [44]; the selected regions contained the laryngeal mucosal lesion and submucosal tissue, and passed the filtering process because of a cell count ≥ 15 [44,45]. A histological score (1 × percentage of cells positive for low brown chromogen intensity) + (2 × percentage of cells positive for medium brown chromogen intensity) + (3 × percentage of cells positive for high brown chromogen intensity) [46] was calculated for cellular staining by the scientific team members (T.-C.C. and C.-G.H.) without knowing the clinical information [36,37].

### 2.6. Systematic Review

#### 2.6.1. Defining the Clinical Question

Systematic searches were performed with the Preferred Reporting Items for Systematic Reviews guidelines (PRISMA) [47] to identify studies on the association between *H. pylori* infection and laryngeal cancer in adults. The clinical question was: Can laryngeal *H. pylori* infection increase the risk of laryngeal cancer?

#### 2.6.2. Identification of Evidence

Electronic databases including PubMed (from 1966 to 14 February 2021) and Ovid^®^ (from 1975 to 14 February 2021) were searched in February 2021 using specific keywords. Abstracts published in English were included. A detailed search using (“*Helicobacter pylori*” [Mesh] or “*(H. pylori)*” [Mesh] or “*Helicobacter* Infections” [Mesh] or “*pylori*” [Mesh]) and (“laryngeal cancer” [Mesh] or “laryngeal squamous cell carcinoma” [Mesh] or “larynx” [Mesh]) was performed. A total of 719 articles were retrieved (PubMed: 516; Ovid^®^: 203). Study references were collected in Endnote x9.3.3 (Clarivate Analytics, Boston, MA, USA).

#### 2.6.3. Study Selection

Primary screening of the articles based on their titles and abstracts was independently performed by two reviewers (H.-H.C. and L.-A.L.) who were blinded to each other’s decisions. Studies were considered eligible for inclusion if they related to the effect of *H. pylori* infection on the development of laryngeal cancer in adulthood patients in randomized-controlled, case-controlled, retrospective studies, or case series. The exclusion criteria were as follows: (1) papers not written in English, (2) research not involving humans (e.g., in vitro or animal research), and (3) research including patients with secondary or recurrent laryngeal cancer. The selection or rejection of articles that could not be decided through primary screening was retained for secondary screening. The results of the two reviewers were cross-checked, and a dataset for secondary screening was prepared, which included 27 full-text articles. Each of the two reviewers (H.-H.C. and L.-A.L.) read all full-text articles and performed the secondary screening separately. Articles that included evaluation of the effect of *H. pylori* infection on the development of laryngeal cancer in adults were selected independently, and the results of the two reviewers were cross-checked. A third person (L.-J.H.) was invited to make a final decision when the two reviewers could not decide whether to accept or reject the article after discussion (*n* = 1).

#### 2.6.4. Data Extraction

The data from each study were extracted by two reviewers (H.-H.C. and L.-A.L.) separately using a data extraction form that was designed in advance. The following information was extracted: first author, year of publication, study design, diagnostic tests of *H. pylori* infection, and case numbers. When the two reviewers had disparities, a third person (L.-J.H.) was invited to make a final decision (*n* = 1).

#### 2.6.5. Study Quality Assessment

Study quality was assessed using the Oxford levels of evidence [48]: level 1: local and current random sample surgery; level 2: systemic review of surveys that allow matching to local circumstances; level 3: local non-random sample; and level 4: case-series.

### 2.7. Statistical Analysis

Continuous variables were summarized as means and standard deviations (SDs), and categorical variables were presented as numbers and percentages. One-way analysis of variance with post-hoc Tukey’s honestly significant difference test was used to compare continuous variables, and the chi-square test was used to compare categorical variables in different groups. Multivariate logistic regression models were used to adjust the association between *H. pylori* infection, LPR, and the development of laryngeal cancer. All p-values were two-sided, and statistical significance was accepted at *p* < 0.05. All statistical analyses were performed using G*Power 3.1.9.2 software (Heinrich-Heine University, Dusseldorf, Germany) and SPSS software (version 25; International Business Machines Corp., Armonk, NY, USA).

## 3. Results

### 3.1. Patient Demographics and Tumor Staging

Ninety-three (89%) men and twelve (11%) women with a mean age of 58.6 ± 14.0 years were enrolled (Table 1). Of the 105 patients, 37 (35%) had VP, 22 (21%) had VFL, and 46 (44%) had laryngeal cancer. We divided the patients into three subgroups according to the final pathology: the VP group, the VFL group, and the laryngeal cancer group. The mean age of the laryngeal cancer group (66.6 ± 11.1 years) was significantly higher than the VP (50.5 ± 11.1 years) and VFL (55.5 ± 15.2 years) groups (*p* < 0.001). The mean RSI was significantly different among the three subgroups (*p* = 0.04); notably, the mean RSI in the patients with VP (8.1 ± 5.3) was significantly higher than that in the patients with laryngeal cancer (5.6 ± 3.8) (*p* = 0.03). There were no significant differences in sex, cigarette smoking, and alcohol consumption among the three groups. Most of the patients with laryngeal cancer were at an early stage (T1 and T2, 49% and 32%, respectively) without neck lymph node metastasis (96% in the N0 stage).

### 3.2. H. pylori Infection Status of the Larynx

Both the PCR and IHC approaches detected a presence of *H. pylori* in the laryngeal lesions (Table 2). One patient with VP had the presence of *ureC* (Figure 2a) which had been confirmed to be *H. pylori* Puno 135 by DNA sequencing (Figure 2b). Therefore, the *H. pylori* DNA positive rates were low (VP: 3%, VFL: 0%, laryngeal cancer: 0%) (*p* = 0.40), and the *H. pylori* antigen-positive rates were not high either (VP: 3% [Figure 1c], VFL: 0%, laryngeal cancer: 2% [Figure 1d]) (*p* = 0.75). The differences between subgroups were not statistically significant.

### 3.3. Expression of E-cadherin and CD1d of the Larynx

In the IHC analysis of *E-cadherin*, 32 (30%) specimens were not included for further comparisons due to an insufficient amount (*n* = 29 [28%]) and low staining quality (*n* = 3 [2%]) (Table 2). Therefore, 73 (70%) patients were included for further statistical analysis. Although the expression of *E-cadherin* in the VFL and laryngeal cancer were higher than that in the VP, the difference did not reach a statistical significance (*p* = 0.08).

In the IHC analysis of *CD1d*, 21 (20%) specimens were not included for further comparisons due to an insufficient amount (*n* = 19 [18%]) and low staining quality (*n* = 2 [2%]). Therefore, 84 (80%) patients were included for further statistical analysis. The expression of *CD1d* in the laryngeal lesions was not statistically significantly different among the three subgroups (*p* = 0.18).

### 3.4. Relationship between H. pylori Status, LPR Symptom, and Related Biomarkers and Malignant Potential and Pathological Status

In this study, we defined benign VP as ‘low malignant potential’, premalignant VFL as ‘intermediate malignant potential’, and laryngeal cancer as ‘malignancy (high malignant potential)’. In the entire cohort (Table 3), the low RSI score was significantly related to high malignant potential (*r* = −0.22; *p* = 0.03), whereas *H. pylori* status of the larynx and related biomarkers were not correlated with malignant potential. In the patients with laryngeal cancer, *H. pylori* status, RSI score, and related biomarkers were not associated with tumor status and neck lymph node status.

### 3.5. Associations of H. pylori Status, RSI Score, and Related Biomarkers with the Risk of Laryngeal Cancer

In the univariate logistic regression model, age (odds ratio, 1.10; 95% confidence interval, 1.06–1.16; *p* < 0.001) and RSI score (odds ratio, 0.90; 95% confidence interval, 0.82–0.996; *p* = 0.04) were significant risk factors for primary laryngeal cancer (Table 4). However, the association of RSI with the risk of laryngeal cancer did not persist after the adjustments for age, male sex, cigarette smoking, and alcohol consumption (odds ratio, 0.96; 95% confidence interval, 0.86–1.07; *p* = 0.46).

### 3.6. Systematic Review

#### 3.6.1. Study Selection and Characteristics of Included Studies

The electronic database search revealed 15 original articles that investigated laryngeal *H. pylori* infection in patients with laryngeal cancer [12,16,20,21,49,50,51,52,53,54,55,56,57,58,59]. The article selection algorithm is shown in Figure 3. The searches of PubMed and Ovid^®^ using the defined terms yielded a total of 719 records. After 625 duplicates were removed, 94 records remained for the title and abstract screen. A total of 54 records were excluded based on article type (reviews, case reports, irrelevant experiments, incomplete data). After full-text assessments of the 40 remaining records, we finally identified and reviewed 15 studies [12,16,20,21,49,50,51,52,53,54,55,56,57,58,59] consisting of nine case-controlled studies and six case series with sufficient data related to laryngeal *H. pylori* infection in patients with laryngeal cancer. Characteristics of the 15 studies are summarized in Table 5. They were published between 2005 and 2018. Out of them, ten reports used PCR, four used the histological method, two used RUT, and two used IHC to diagnose laryngeal *H. pylori* infection. We extracted the data of 749 patients with laryngeal cancer and 375 controls.

#### 3.6.2. Level of Evidence

Out of the 15 included studies, nine studies provided level 3 evidence, and six provided level 4 evidence, indicating a relatively low level of evidence.

#### 3.6.3. Study Selection and Characteristics of Included Studies

Including our patients, 234 (29.4%) of the total 795 patients with laryngeal cancer had laryngeal *H. pylori* infection, whereas 69 (16.7%) of the total 412 controls had laryngeal *H. pylori* infection. Wide variabilities in prevalence from 0% to 81% in patients with laryngeal cancer and from 0% to 46% in controls were observed.

Furthermore, 227 (40.0%) of 567 patients with laryngeal cancer and 68 (25.9%) of controls had laryngeal *H. pylori* infection, assessed by PCR [12,16,20,51,52,53,55,56,57,58]. Fourteen (4.3%) of 325 patients with laryngeal cancer and two (1.1%) of controls had laryngeal *H. pylori* infection, assessed by RUT or IHC or histopathology [12,21,49,50,54,59].

Of the fresh or fresh-frozen samples, 223 (49.7%) of 449 patients with laryngeal cancer and 67 (25.5%) of 263 controls had laryngeal *H. pylori* infection [20,49,50,51,52,53,54,55,56,58,59]. Of the FFPE samples, ten (2.6%) of 390 patients with laryngeal cancer and two (1.1%) of 17 controls had laryngeal *H. pylori* infection [12,16,21,57,59].

## 4. Discussion

The most recent meta-analysis on the association between *H. pylori* infection and carcinoma of the larynx and pharynx showed that *H. pylori* infection, diagnosed by PCR or enzyme-linked immunosorbent assay (ELISA), is related to laryngeal cancer, especially in the hospital-based control group [17]. However, Zou et al. also considered that ELISA may be less specific for the detection of *H. pylori* than PCR or generate false-positive results [17]. A case-controlled study in the U.S. with 119 patients with laryngeal cancer and 111 controls also reported no significant association between *H. pylori* seropositivity and laryngeal cancer [60]. In addition, another case-controlled study including 31 patients with laryngeal cancer and 28 benign controls also showed no statistically significant differences in serology between cancer and benign groups [21]. In this hospital-based study, we used three approaches to investigate the relationship between laryngeal *H. pylori* infection and laryngeal cancer.

First, we tested the infection with PCR, which is known to have high sensitivity [61]. We also examined for colonization with IHC assays, which can sensitively locate the colonies [62] of *H. pylori*. However, the *H. pylori* positivity rates in both PCR and IHC assays among the laryngeal cancer and non-cancer subgroups were relatively low.


Second, for the disease characteristic related to reflux, we used the RSI questionnaire to identify patients with a higher propensity for LPR caused by *H. pylori* infection. Interestingly, RSI in the VP group was significantly higher than those in the VFL and laryngeal cancer groups. After adjusting traditional risk factors of laryngeal cancer, the RSI and LPR were no longer associated with the risk of laryngeal cancer. Thus, the lack of significance in the LPR profile among the subgroups further suggested that persistent *H. pylori* infection and LPR might not always be mandatory for the development of laryngeal cancer.

Third, the expression of the *H. pylori*- and LPR-related biomarkers (*E-cadherin* and *CD1d*) were measured in more than 70% of our study patients. According to in vitro studies, the protein expression of *E-cadherin* can be down-regulated by *H. pylori* via methylation of *E-cadherin* promoter [63,64,65]. We found that the expression of *E-cadherin* was not related to the risk and pathological status of laryngeal cancer. Furthermore, cholesteryl phosphatidyl α-glucoside in *H. pylori* cell wall has been shown to bind to CD1d [66] and then can be recognized by natural killer T cells [67,68]. Besides, the CD1d-natural killer T cell axis is involved in the immunological response to LPR in humans [33]. However, from our data, the expression of *CD1d* was not significantly different among groups. The results did not support the pathogenetic effect of *H. pylori* and LPR on the risk of laryngeal cancer in Taiwanese patients.

Based on the three aforementioned approaches, our results demonstrate no correlations between laryngeal *H. pylori* infection and laryngeal cancer. Possible explanations include: (1) Taiwanese patients are less susceptible to *H. pylori*, and it is harder for *H. pylori* to migrate from gastric mucosa to the larynx; (2) persistent laryngeal *H. pylori* infection might not be necessary for the development of malignant lesions in Taiwanese patients. Our findings suggest that a connection between *H. pylori* infection and the pathogenesis of laryngeal cancer in Taiwanese patients is not very likely.

The reported prevalence of *H. pylori* infection and laryngeal cancer varies widely between studies (ranging from 0 to 81%), depending on the area researched and the diagnostic method used [12,16,20,21,49,50,51,52,53,54,55,56,57,58,59] (Table 5). Because of the heterogeneity in sample types, detection methods, and study designs of previous reports, it was difficult to conduct a multi-country meta-analysis to determine the effect of laryngeal *H. pylori* infection on the development of laryngeal cancer. We performed a systematic review in this study, which indicated that laryngeal *H. pylori* infection was associated with laryngeal cancer worldwide with a low level of evidence.

There were some interesting findings from the systematic review to be considered. First, the low prevalence (4.3%) of laryngeal *H. pylori* infection of the laryngeal malignancy, determined by a RUT, IHC, or histopathology, seemed not to support the role of *H. pylori* in the development of laryngeal cancer [12,21,49,50,54,59]. In contrast, laryngeal *H. Pylori* infection detected by PCR assay was associated with a “higher than expected” prevalence (40.0%) in patients with laryngeal cancer than that (25.9%) in controls [12,16,20,51,52,53,55,56,57,58]. Second, the FFPE samples of laryngeal cancer might be less suitable for detecting *H. pylori* infection than the fresh or fresh-frozen samples, because the obviously low detecting rate (2.6%) of the FFPE sample [12,16,21,57,59] comparing with that (49.7%) of fresh or fresh-frozen samples [20,49,50,51,52,53,54,55,56,58,59]. Notably, *H. pylori* infection in the laryngeal mucosa may play a role in the development of laryngeal cancer after destroying the mucosal lining of the larynx [20]. Therefore, to detect laryngeal *H. pylori* infection, examining fresh or fresh-frozen laryngeal samples with PCR assay may be the most optimal approach for patients with laryngeal lesions in order to stratify the risk of laryngeal cancer.

The primary focus of this study was to detect *H. pylori* infection in laryngeal tissues. No patient in the cohort had a serology test for *H. pylori* and 41 patients had data of gastric biopsy from medical chart reviews. Therefore, we could not determine whether the patient was gastric *H. pylori*-naïve, currently *H. pylori*-infected, or with *H. pylori* past infection. Because cancer cells often drive out *H. pylori* from the tissue during the “hit-and-run” carcinogenesis process [69], *H. pylori* may induce LPR in the beginning stage of malignant transformation [17] and disappear during the development of laryngeal cancer. It is hard to conclude that the lack of *H. pylori* in the laryngeal tissue is equal to a naïve stomach; therefore, the lack of *H. pylori* in laryngeal cancer could not be arbitrarily interpreted as a lack of contribution from *H. pylori* on the development of laryngeal cancer. The relationships between gastric *H. pylori* infection and laryngeal cancer are still inconclusive [12,13,14,15,16].

The current study took a comprehensive approach to identifying *H. pylori* infection in various types of laryngeal lesions, and the results demonstrated a lack of *H. pylori*-associated changes in RSI score, *E-cadherin* expression, and *CD1d* expression of the larynx among patients with newly diagnosed laryngeal cancer. However, there were still some limitations to this study. First, the sample size of the case series was not very big, and there were no *H. pylori*-PCR positive cases in the laryngeal cancer group, which might have resulted in a lack of statistical power. Second, cross-sectional studies are not able to determine the causal effect relationship between *H. pylori* and laryngeal cancer [70]. Third, this study used only one *ureC* gene to confirm *H. pylori* infection, and there are some other existing PCR methods that might be more effective in detecting *H. pylori*, such as *cagA*, *vacA* or 16s rRNA [71,72]. Lastly, this study focused on the presence of *H. pylori* in the larynx and laryngeal cancer without examination of gastric and oral *H. pylori* infection. Further longitudinal studies with a larger sample size are warranted to fully establish the role of *H. pylori* infection in the stomach and oral cavity among patients with laryngeal cancer.

## 5. Conclusions

To the best of our knowledge, this was the most comprehensive analysis to date of the clinical and histological manifestations of *H. pylori* infection in both benign and malignant laryngeal lesions. Laryngeal *H. pylori* infections were relatively rare, and their role in the development of laryngeal cancer might not be evident in Taiwan. However, our systematic review supported that laryngeal *H. pylori* infection could be a risk factor for laryngeal cancer globally. Furthermore, detecting laryngeal *H. pylori* infection using PCR assays in the fresh or fresh-frozen samples seemed to be more optimal than other approaches. Despite we only included patients with the newly diagnosed laryngeal disease, the effects of laryngeal *H. pylori* infection on the different disease courses of laryngeal cancer need to be investigated in future studies.

## Figures and Tables

**Figure 1 microorganisms-09-01129-f001:**
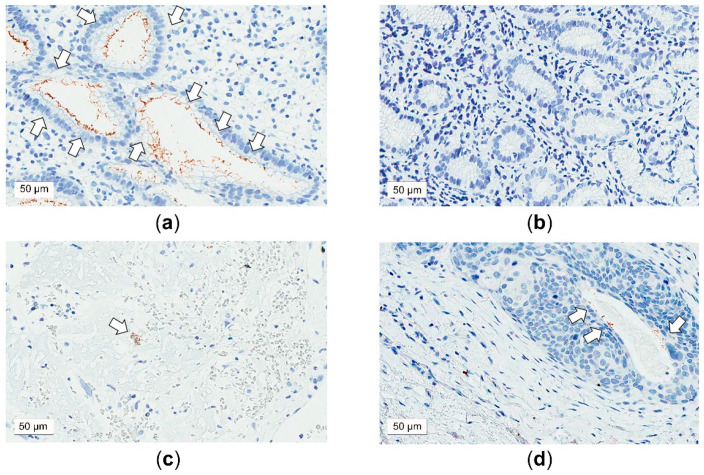
Representative examples of *Helicobacter pylori* (*H. pylori*) antigen detection using immunohistochemistry (IHC). (**a**) Positive IHC staining of *H**. pylori* antigens (brown color) from a gastric biopsy in a patient with gastric adenocarcinoma and *H**. pylori* infection (arrows). (**b**) Negative IHC staining in the gastric tissue of a patient with *H**. pylori*-negative gastric adenocarcinoma. (**c**) Positive IHC staining of *H**. pylori* antigen (brown color) in the laryngeal tissue of a patient with vocal polyp (arrow). (**d**) Positive IHC staining of *H**. pylori* antigen (brown color) in the laryngeal tissue of a patient with laryngeal cancer (arrows). Original magnification: 40×.

**Figure 2 microorganisms-09-01129-f002:**
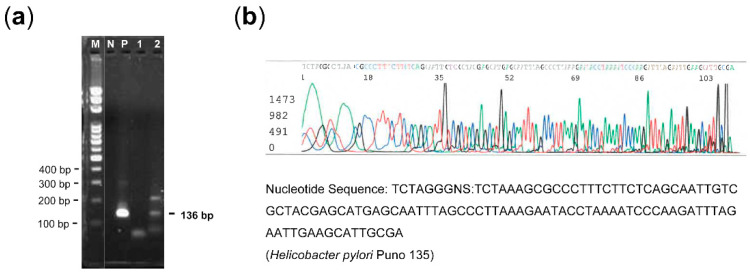
*Helicobacter pylori* DNA detection. (**a**) Gel electrophoresis of polymerase chain reaction product of *ureC* (136 bp) gene. (**b**) DNA sequencing for the presence of *ureC* in one patient with vocal polyp and laryngeal *Helicobacter pylori* infection (genotype: Puno135). Lane 2: positive *ureC*; Lane 1: negative *ureC*; P: positive control; N: negative control; M: 100-bp DNA marker.

**Figure 3 microorganisms-09-01129-f003:**
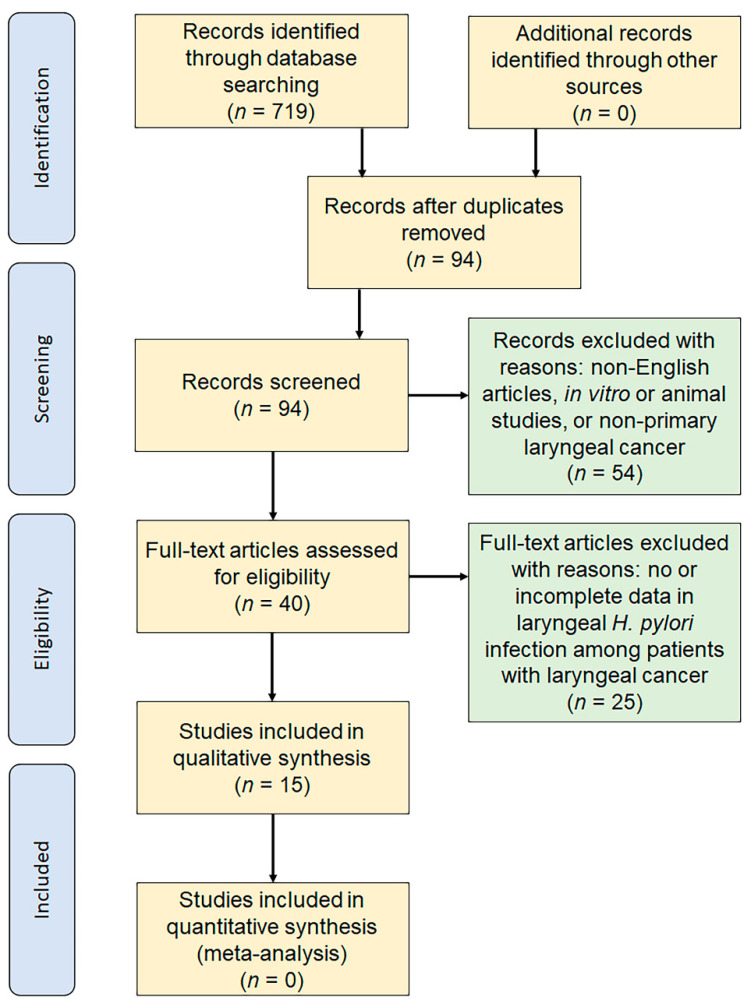
The flow of study identification, inclusion, and exclusion.

**Table 1 microorganisms-09-01129-t001:** Clinicopathological characteristics of the entire cohort and subgroups.

Variables	All	VP	VFL	Laryngeal Cancer	*p*-Value ^1^
Patients	(*n* = 105)	(*n* = 37)	(*n* = 22)	(*n* = 46)	
Clinical characteristics					
Male sex, *n* (%)	93 (89)	30 (81)	20 (91)	43 (94)	0.20
Age (years), mean (SD)	58.6 (14.0)	50.5 (11.1) ^2^	55.5 (15.2) ^3^	66.6 (11.1) ^2,3^	<0.001
Cigarette smoking, *n* (%)	82 (78)	24 (65)	19 (86)	39 (85)	0.05
Alcohol consumption, *n* (%)	67 (64)	19 (51)	17 (77)	31 (67)	0.11
RSI score, mean (SD)	5.9 (3.8)	8.1 (5.3) ^2^	6.4 (3.7)	5.6 (3.8) ^2^	0.04
Pathological characteristics					
Tis, *n* (%)	‒	‒	‒	3 (6)	**‒**
T1, *n* (%)	‒	‒	‒	23 (49)	**‒**
T2, *n* (%)	‒	‒	‒	15 (32)	**‒**
T3, *n* (%)	‒	‒	‒	3 (6)	**‒**
T4, *n* (%)	‒	‒	‒	3 (6)	**‒**
N0, *n* (%)	‒	‒	‒	45 (96)	**‒**
N1, *n* (%)	‒	‒	‒	1 (2)	**‒**
N2, *n* (%)	‒	‒	‒	1 (2)	**‒**

Abbreviations: RSI: Reflux Symptom Index; SD: standard deviation; VFL: vocal fold leukoplakia; VP: vocal polyp. ^1^ Data were compared using one-way analysis of variance with post-hoc Tukey’s honestly significant difference tests for continuous variables, and the chi-square test for categorical variables. ^2^
*p*-Value < 0.05 when the variable in the VP subgroup was compared with the VFL or laryngeal cancer subgroup. ^3^
*p*-Value < 0.05 when the variable in the VFL subgroup was compared with the laryngeal cancer subgroup. Significant *p*-values are marked in bold.

**Table 2 microorganisms-09-01129-t002:** Helicobacter *pylori* status, LPR, and related biomarkers in the entire cohort and subgroups.

Variables	All	VP	VFL	Laryngeal Cancer	*p*-Value ^1^
*Helicobacter pylori* status in the laryngeal lesions (*n* = 105)					
Patent number (%)	105 (100)	37 (35)	22 (21)	46 (44)	
DNA positivity, *n* (%)	1 (1)	1 (3)	0 (0)	0 (0)	0.40
Antigen positivity, *n* (%)	2 (2)	1 (3)	0 (0)	1 (2)	0.75
Immunohistochemistry of *Helicobacter pylori*- and LPR-related biomarkers (*n* = 84)					
Patent number (%)	73 (100)	24 (33)	17 (23)	32 (44)	
Expression of *E-cadherin*, mean (SD)	95.5 (71.1)	69.8 (55.8)	115.0 (65.2)	104.4 (80.3)	0.08
Patent number (%)	84 (100)	26 (31)	17 (20)	41 (49)	
Expression of *CD1d*, mean (SD)	41.8 (48.6)	23.0 (35.6)	37.7 (41.7)	53.0 (60.2)	0.18

Abbreviations: RSI: Reflux Symptom Index; SD: standard deviation; VFL: vocal fold leukoplakia; VP: vocal polyp. ^1^ Data were compared using one-way analysis of variance with post-hoc Tukey’s honestly significant difference tests for continuous variables, and the chi-square test for categorical variables.

**Table 3 microorganisms-09-01129-t003:** Spearman correlations of *Helicobacter pylori* status, RSI score, and related characteristics with malignant potential and pathological status.

Variables	Malignant Potential	Tumor Status	Neck Lymph Node Status
*Helicobacter pylori* DNA positivity	*r* = −0.12; *p* = 0.23	‒	‒
	*n* = 105	*n* = 46	*n* = 46
*Helicobacter pylori* antigen positivity	*r* = −0.01; *p* = 0.91	*r* = 0.08; *p* = 0.61	*r* = −0.05; *p* = 0.76
	*n* = 105	*n* = 46	*n* = 46
RSI score	*r* = −0.22; *p* = 0.03	*r* = 0.20; *p* = 0.19	*r* = 0.02; *p* = 0.89
	*n* = 105	*n* = 46	*n* = 46
E-cadherin	*r* = 0.17; *p* = 0.14	*r* = 0.15; *p* = 0.43	*r* = −0.17; *p* = 0.28
	*n* = 73	*n* = 32	*n* = 32
CD1d	*r* = 0.11; *p* = 0.31	*r* = 0.02; *p* = 0.92	*r* = −0.17; *p* = 0.28
	*n* = 84	*n* = 41	*n* = 41

Abbreviations: LPR: laryngopharyngeal reflux; RSI: Reflux Symptom Index. Significant *p*-values are marked in bold.

**Table 4 microorganisms-09-01129-t004:** Predictive values of variables of interest for the risk of primary laryngeal cancer.

Variables	Odds Ratio (95% CI) ^1^	*p*-Value	Odds Ratio (95% CI) ^2^	*p*-Value
Model	Univariate analysis		Multivariate analysis	
Age	1.10 (1.06–1.15)	<0.001	1.11 (1.06–1.16)	**<0.001**
Male sex	2.58 (0.66–10.14)	0.18	0.84 (0.14–4.98)	0.85
Cigarette smoking	2.07 (0.77–5.57)	0.15	2.40 (0.66–8.70)	0.18
Alcohol consumption	1.32 (0.59–2.96)	0.50	0.92 (0.34–2.53)	0.87
*Helicobacter pylori* DNA positivity	<0.001 (<0.001–)	>0.99	<0.001 (<0.001–)	>0.99
*Helicobacter pylori* antigen positivity	1.29 (0.08–21.18)	0.86	0.46 (0.03–8.13)	0.60
RSI score	0.90 (0.82–0.996)	0.04	0.96 (0.86–0.1.07)	0.46
Expression of *E-cadherin*	1.00 (1.00–1.01)	0.34	1.00 (0.99–1.01)	0.75
Expression of *CD1d*	1.01 (1.00–1.02)	0.08	1.00 (0.99–1.01)	0.59

Abbreviations: CI: confidence interval; RSI: Reflux Symptom Index; SD: standard deviation; VFL: vocal fold leukoplakia; VP: vocal polyp. ^1^ The univariate logistic regression model. ^2^ The multivariate logistic regression model with adjustment for age, male sex, cigarette smoking, and/or alcohol consumption. Significant *p*-values are marked in bold.

**Table 5 microorganisms-09-01129-t005:** Report characteristics and *H. pylori* infection rates in the laryngeal tissues (in chronological order).

First Author (Year)	Country	Sample Type	Detection Methods	Study Design	*H. pylori* Infection Rate in the Laryngeal Lesions			Level of Evidence
					Laryngeal Cancer	Controls	*p*-Value	
Akbayir N (2005) [49]	Turkey	FFPE	H/IHC	CCS	0% (0/50)	0% (0/50)	>0.99	3
Kizilay A (2006) [50]	Turkey	FFPE	H	CCS	0% (0/90)	0% (0/30)	>0.99	3
Titiz A (2008) [51]	Turkey	FF	PCR	CCS	81% (17/21)	0% (0/19)	<0.001	3
Masoud N (2008) [59]	Iran	F/FFPE	RUT/H	CCS	0% (0/44)	0% (0/30)	>0.99	3
Grbesa I (2008) [52]	Croatia	FF	PCR	CS	26% (9/35)	NA	NA	4
Shi Y (2011) [53]	China	FF	PCR	CCS	76% (45/59)	32% (13/41)	<0.001	3
Gong H (2012) [20]	China	FF	PCR	CCS	72% (58/81)	25% (19/75)	<0.001	3
Siupsinskiene N (2013) [54]	United States	F	RUT	CS	46% (6/13)	9% (1/11)	0.047	4
Burduk PK (2013) [55]	Poland	F	PCR	CS	47% (35/75)	NA	NA	4
Genç R (2013) [21]	Turkey	FFPE	IHC	CCS	0% (0/31)	0% (0/28)	>0.99	3
Fellmann J (2014) [16]	Switzerland	FFPE	PCR	CS	50% (2/4)	25% (1/4)	0.47	4
Amizadeh M (2015) [56]	Iran	FF	PCR	CCS	33% (24/72)	46% (33/72)	0.13	3
Yilmaz I (2016) [57]	Turkey	FFPE	PCR	CS	1% (1/74)	NA	NA	4
Barakat G (2016) [58]	Egypt	F	PCR	CCS	59% (29/49)	7% (1/15)	<0.001	3
Pajić Matić I (2018) [12]	Croatia	FFPE	H/PCR	CS	14% (7/51)	NA	NA	4
This study (2021)	Taiwan	FFPE	PCR/IHC	CS	0% (0/46)/2% (1/46)	3% (1/37)/3% (1/37)	0.26	4
Total ^1^					29.4% (234/795)	16.7% (69/412)		

Abbreviations: CCS: case-controlled study; CS: case-series; F, fresh samples; FF, fresh-frozen samples; FFPE, formalin-fixed paraffin-embedded samples; H, histopathology; *H. pylori*, *Helicobacter pylori*; IHC, immunocytochemistry; NA, not available; PCR, polymerase chain reaction; RUT, rapid urease test. ^1^ Result of a pooled data analysis. Significant *p*-values are marked in bold.

## Data Availability

The data presented in this study are available on request from the corresponding author. The data are not publicly available due to ethical restrictions.

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
