# Peer review of "Laryngeal Helicobacter pylori Infection and Laryngeal Cancer-Case Series and a Systematic Review"

_microorganisms, 2021, doi:10.3390/microorganisms9061129_

Round 1

Reviewer 1 Report

The study is an interesting and suggest that laryngeal H. pylori infection may not be the etiological factor of laryngeal cancer in Taiwan.

I find this study interesting, the results are well presented and deserving publication although I have some comments for the consideration of the authors.

Abstract section:

The aim is not clear. Please, you should better explain the aim in abstract section.

Introduction section:

- The authors should report the importance of possible coinfection with others microorganisms such as EBV as reported in:

- HELICOBACTER PYLORI AND EPSTEIN–BARR CO-INFECTION IN GASTRIC DISEASE. Fasciana T., Capra G., Calà C., Zambuto S, Mascarella C., Colomba C., Di Carlo P., Giammanco A. PhOL  2017

- Helicobacter pylori and Epstein-Barr virus infection in gastric diseases: Correlation with IL-10 and IL1RN polymorphism. Journal of Oncology Volume 2019.

The conclusions are intuitive and the findings don't support an innovative discovery.  Please, insert some comments to support the aims of study.

Author Response

Reviewer 1’s comments:

The study is an interesting and suggest that laryngeal H. pylori infection may not be the etiological factor of laryngeal cancer in Taiwan.

I find this study interesting, the results are well presented and deserving publication although I have some comments for the consideration of the authors.

REPLY. Thank you for helping us improve our work. Your comments are genuinely appreciated. We’ve revised the manuscript accordingly, and hopefully it will meet your expectations.

Abstract section:

The aim is not clear. Please, you should better explain the aim in abstract section.

REPLY. Thank you for pointing this out. We have rewritten the abstract to better present our study aims.

Modified text, Page 1, Line 24-28

Helicobacter pylori (H. pylori) infection has been demonstrated in patients with laryngeal cancer. However, whether H. pylori is a risk factor for laryngeal cancer is not conclusive. This study aimed to investigate the possible association of laryngeal H. pylori infection in patients with newly diagnosed laryngeal cancer and perform a systematic review of previous reports about H. pylori infection laryngeal cancer. …’

-->

Helicobacter pylori (H. pylori) infection involves the development of gastric cancer and may be associated with laryngeal cancer. However, laryngeal H. pylori infection in Taiwanese patients with newly diagnosed laryngeal cancer has not been reported. This study was aimed to investigate the possible association between laryngeal H. pylori infection and laryngeal cancer in Taiwan and perform a systematic review of previous reports in other countries. …’

Introduction section:

- The authors should report the importance of possible coinfection with others microorganisms such as EBV as reported in:

- HELICOBACTER PYLORI AND EPSTEIN–BARR CO-INFECTION IN GASTRIC DISEASE. Fasciana T., Capra G., Calà C., Zambuto S, Mascarella C., Colomba C., Di Carlo P., Giammanco A. PhOL  2017

- Helicobacter pylori and Epstein-Barr virus infection in gastric diseases: Correlation with IL-10 and IL1RN polymorphism. Journal of Oncology Volume 2019.

REPLY. Thank you very much for this cogent comment. We have added this information in the introduction in the revised manuscript.

Modified text, Page 2, Line 53-56

‘… cancer among East Asian populations [7].’

-->

‘… cancer among East Asian populations [8]. Furthermore, H. pylori and Epstein–Barr virus co-infection can be involved in the development of gastric cancer [9,10].’

The conclusions are intuitive and the findings don't support an innovative discovery. Please, insert some comments to support the aims of study.

REPLY. Thank you very much for these cogent comments. We have amended the conclusions in the revised manuscript.

Modified text, Page 13, Line 460-469

‘To the best of our knowledge, this is the most comprehensive analysis to date of the clinical and histological manifestations of H. pylori infection and LPR in both benign and malignant laryngeal lesions. We found that laryngeal H. pylori infection and LPR were relatively rare among our patients using PCR and IHC examinations. The involvement of H. pylori infection, LPR, and related characteristics in the pathogenesis of laryngeal cancer might not be evident in this study. However, our systematic review still supported laryngeal H. pylori infection can be a risk factor for laryngeal cancer. Despite we only included patients with the newly diagnosed laryngeal disease, the effects of laryngeal H. pylori infection and LPR between the different disease courses of laryngeal cancer need to be investigated in future studies.’

-->

‘To the best of our knowledge, this was the most comprehensive analysis to date of the clinical and histological manifestations of H. pylori infection in both benign and malignant laryngeal lesions. Laryngeal H. pylori infection were relatively rare, and their role in the development of laryngeal cancer might not be evident in Taiwan. However, our systematic review supported laryngeal H. pylori infection could be a risk factor for laryngeal cancer globally. Furthermore, detecting laryngeal H. pylori infection using PCR assays in the fresh or fresh-frozen samples seemed to be more optimal than other approaches. Despite we only included patients with the newly diagnosed laryngeal disease, the effects of laryngeal H. pylori infection on the different disease courses of laryngeal cancer need to be investigated in future studies.’

Reviewer 2 Report

Minor issues

Figure 2 - part ‘Screening’ it is unclear how ‘Records identified through database searching (n=54)’ were selected and how does this step differs from the previous ‘Records identified through database searching (n=719)’?

Table 5: the fifth line from the bottom - there is ‘Switzerland’ instead of ‘Zurich’

  • abbreviations below - There is ‘FFPR’ instead of’ FFPE’.

Line 374: (2) - this point does not seem to be an explanation.

Author Response

Reviewer 2’s comments:

Minor issues

Figure 2 - part ‘Screening’ it is unclear how ‘Records identified through database searching (n=54)’ were selected and how does this step differs from the previous ‘Records identified through database searching (n=719)’?

REPLY. We appreciate your question. The 54 records were excluded due to non-English articles, in vitro or animal studies, or non-primary laryngeal cancer (n = 54). We have amended Figure 2 to better clarify the study flow. Thank you very much for helping us improve our work.

Modified text, Page 10, lines 357-358

Figure 2. The flow of study identification, inclusion, and exclusion.’

--> Modified Figure 2 as indicated.

Table 5: the fifth line from the bottom - there is ‘Switzerland’ instead of ‘Zurich’

REPLY. Thank you for pointing this out. We have revised the Table 5.

Modified text, Page 11, Table 5

Fellmann J (2014) [14]

Zurich

FFPE

PCR

CS

50% (2/4)

25% (1/4)

0.47

4

-->

Fellmann J (2014) [14]

Switzerland

FFPE

PCR

CS

50% (2/4)

25% (1/4)

0.47

4

  • abbreviations below - There is ‘FFPR’ instead of’ FFPE’.

REPLY. We appreciate this cogent comment. We have amended the revised Table 5. Thanks!

Modified text, Page 11, Table 5

‘Abbreviations: CCS: case-controlled study; CS: case-series; F, fresh samples; FF, fresh-frozen samples; FFPR, formalin-fixed paraffin-embedded’

à

‘Abbreviations: CCS: case-controlled study; CS: case-series; F, fresh samples; FF, fresh-frozen samples; FFPE, formalin-fixed paraffin-embedded’

Line 374: (2) - this point does not seem to be an explanation.

REPLY. We appreciate this in-depth comment. We have deleted this point due to redundancy. The quality of this revised manuscript has been substantially improved. Thank you very much!

Modified text, Page 12, lines 404-406

‘… (1) Taiwanese patients are less susceptible to H. pylori, and that it is harder for H. pylori to migrate from gastric mucosa to the larynx; (2) Interactions between laryngeal H. pylori infection and host factors did not seem to be associated with laryngeal cancer; (3) Persistent laryngeal H. pylori infection might be not necessary for the …’

-->

‘… (1) Taiwanese patients are less susceptible to H. pylori, and that it is harder for H. pylori to migrate from gastric mucosa to the larynx; (2) Persistent laryngeal H. pylori infection might be not necessary for the…’

Reviewer 3 Report

The manuscript entitled "Laryngeal Helicobacter pylori Infection and Laryngeal Cancer-Case Series and a Systematic Review" is well written and designed. A good attempt at this study was to analyze patient samples.  This paper can be accepted for publication only after some major changes.

1. The second paragraph of Introduction authors should add  more details about Helicobacter pylori. What is HP, how it can be caused. What are the different factors responsible and associate with it. I would recommend adding some important factors like diet, environment,  Stress, alcohol etc. Author should follow the papers DOI: 10.3390/cancers12040816 and DOI: 10.21203/rs.2.23065/v1 as reference. 

2. Author used only one primer Urease to confirm HP which is not satisfactory. Author should have other genes to confirm the HP infection like CagA , vacA etc. Confirm HP using 16srRNA primers for the authentication. 

3. Describe the rationale to choose  E-cadherin and CD1d markers for this study. 

4. HP generally associated with gastric cancers, what is the preliminary work and hypothesis behind to choose laryngeal cancer for this study. Describe the details, background and strong hypothesis. 

5. How do authors correlate the data to perform meta analysis from the database? 

6. Figure 1 IHC data looks not highly significant. Authors should quantify the data and represent it as a bar diagram. Present the IHC image clearly by arrow showing what is the area where they want to show the difference.  Also mention what is the control they used here. 

7. Add the PCR data as supplementary to support their conclusion. Provide the gel image as supplementary. 

Author Response

Reviewer 3’s comments:

The manuscript entitled "Laryngeal Helicobacter pylori Infection and Laryngeal Cancer-Case Series and a Systematic Review" is well written and designed. A good attempt at this study was to analyze patient samples. This paper can be accepted for publication only after some major changes.

REPLY. We appreciate these encouraging comments. We have amended the modified text concerning your suggestions. The quality of this revised manuscript has been substantially improved. Thank you very much!

  1. The second paragraph of Introduction authors should add more details about Helicobacter pylori. What is HP, how it can be caused. What are the different factors responsible and associate with it. I would recommend adding some important factors like diet, environment, stress, alcohol etc. Author should follow the papers DOI: 10.3390/cancers12040816 and DOI: 10.21203/rs.2.23065/v1 as reference.

REPLY. Thank you very much for these valuable comments. We’ve revised the introduction with the information added.

Modified text, Pages 1-2, lines 47-56

Helicobacter pylori (H. pylori) is a microaerophilic Gram-negative bacterium that lives in the stomach and duodenum. Despite medical advances, the prevalence of H. pylori is high worldwide (33%–50%) [5]. H. pylori is known to play a role in the carcinogenesis of gastric cancer and mucosa-associated lymphoid tissue lymphoma [6]. It is a major risk factor for gastric cancer among East Asian populations [7].’

-->

Helicobacter pylori (H. pylori) is a microaerophilic Gram-negative bacterium that lives in the stomach and duodenum. The helical shape, number of flagella, motility, and urease secretion facilitate H. pylori to survive in the acidic environment of the stomach [5]. Despite medical advances, the prevalence of H. pylori is high worldwide (33%–50%) [6]. H. pylori is known to play a role in the carcinogenesis of gastric cancer and mucosa-associated lymphoid tissue lymphoma [7]. It is a major risk factor for gastric cancer among East Asian populations [8]. Furthermore, H. pylori and Epstein–Barr virus co-infection can be involved in the development of gastric cancer [9,10]. Host genetic factors, alcohol, capsaicin consumption, ingestion of inflammatory foods, and stress levels are known risk factors for gastric cancer [11].’

  1. Author used only one primer Urease to confirm HP which is not satisfactory. Author should have other genes to confirm the HP infection like CagA, vacA etc. Confirm HP using 16srRNA primers for the authentication.

REPLY. Thank you for these in-depth comments. Unfortunately, we did not detect cagA, vacA, and 16s rRNA to confirm H. pylori infection in this study. However, we performed DNA sequencing in the presence of ureC gene to confirm H. pylori. We added this information in the methods and results. Furthermore, we add this important information in the modified study limitations. We appreciate the excellent suggestion.

Modified text, Page 3, lines 146-148

‘… staining with ethidium bromide (0.5 mg/mL).’

-->

‘… staining with ethidium bromide (0.5 mg/mL). In the case of the presence of ureC, DNA sequencing was performed, and the nucleotides of DNA sequences were compared by using the Basic Local Alignment Search Tool (http:// https://blast.ncbi.nlm.nih.gov/).’

Modified text, Page 6, lines 255-258

‘Both the PCR and IHC approach detected a presence of H. pylori in the laryngeal lesions. The H. pylori DNA positive rates were low …’

-->

‘… able to determine the causal effect relationship between H. pylori and laryngeal cancer [70]. Lastly, this study focused on the presence of H. …’

Modified text, Page 13, Lines 451-454

‘… newly diagnosed laryngeal cancer. Second, this study focused on …’

-->

‘… able to determine the causal effect relationship between H. pylori and laryngeal cancer [70]. Third, this study used only one ureC gene to confirm H. pylori infection, and there are some other existing PCR methods that might be more effective in detecting H. pylori, such as cagA, vacA or 16s rRNA [71,72]. Lastly, this study focused on the presence of H. …’

  1. Describe the rationale to choose E-cadherin and CD1d markers for this study.

REPLY. Thank you very much for the valuable comment. We’ve revised the second last paragraph of the introduction, in which the rationale to choose E-cadherin and CD1d markers were elucidated to the readers.

Modified text, Page 2, lines 84-92

‘From a molecular genetic perspective, both H. pylori infection [29] and LPR [30] are associated with a decreased expression of E-cadherin. This can then reduce the apoptosis of HEp-2 cells, which are considered to originate from human laryngeal carcinoma [31]. Furthermore, both H. pylori [32] and LPR [33] have been shown to induce an over-expression of CD1d in mucosal tissue, and both CD1d and lipid antigens can stimulate CD1d-restricted T cells (i.e., natural killer T cells) to further activate innate and adaptive immune cells in the tumor microenvironment [34].’

-->

‘From a molecular genetic perspective, both H. pylori infection [29] and LPR [30] are associated with a decreased expression of E-cadherin. This can then reduce the apoptosis of HEp-2 cells, which are considered to originate from human laryngeal carcinoma [31]. Furthermore, both H. pylori [32] and LPR [33] have been shown to induce an over-expression of CD1d in mucosal tissue, and both CD1d and lipid antigens can stimulate CD1d-restricted T cells (i.e., natural killer T cells) to further activate innate and adaptive immune cells in the tumor microenvironment [34]. Therefore, it would be interesting to investigate the difference in H. pylori- and/or LPR-associated expression of E-cadherin and CD1d between various laryngeal lesions.’

  1. HP generally associated with gastric cancers, what is the preliminary work and hypothesis behind to choose laryngeal cancer for this study. Describe the details, background and strong hypothesis.

REPLY. Thank you very much for these in-depth comments. We’ve revised the introduction, in which we presented the background and hypothesis for choosing to investigate the connection between laryngeal H. pylori infection and laryngeal cancer.

Modified text, Page 2, lines 68-73

‘Increasing evidence supports that H. pylori can play a role in the development of laryngeal cancer, including that (1) H. pylori can live in the normal larynx [10], vocal polyps (VP) [20], vocal fold leukoplakia (VFL) [21], and precancerous lesions [22], and (2) H. pylori can induce systemic inflammation and promote tumorigenesis [23].’

-->

‘Increasing evidence supports that H. pylori may play a role in the development of laryngeal cancer, including that (1) H. pylori can be detected in the normal larynx [16], vocal polyps (VP) [24], vocal fold leukoplakia (VFL) [25], and precancerous lesions [26], and (2) H. pylori can induce systemic inflammation and promote tumorigenesis [26]. A previous systematic review suggested that H. pylori infection of the stomach, larynx or non-specific sites was related to laryngeal cancer [17].’

Modified text, Page 2, lines 93-97

However, the role of laryngeal H. pylori infection in the development of laryngeal cancer (inflamed larynx - lesions) remains to be controversial [30]. The aims of this study were …’

-->

‘The role of gastric H. pylori infection in the pathogenesis of gastric cancer has been substantially studied. Some research has also suggested an association between laryngeal H. pylori infection and laryngeal cancer (inflamed larynx - lesions), however the connection remains to be controversial [35]. We hypothesized that laryngeal H. pylori infection was involved in the development of laryngeal cancer. The aims of this study were …’

  1. How do authors correlate the data to perform meta analysis from the database?

REPLY. Thank you very much for this cogent comment. To our best knowledge, there was no systematic review and meta-analysis aimed to investigate the associated between laryngeal H. pylori infection and the development laryngeal cancer globally. Therefore, we performed a systematic review in this study. Because of the low level of evidence and the heterogeneous sample types, detection methods, and study designs of previous reports, it is difficult to perform a meta-analysis to determine the effect of laryngeal H. pylori infection on the development of laryngeal cancer globally. However, we found that the FFPE samples of laryngeal cancer might be less suitable for detecting H. pylori infection than the fresh or fresh-frozen samples because the obviously low detecting rate (2.6%) of the FFPE sample [12,16,21,57,59] comparing with that (49.7%) of fresh or fresh-frozen samples [20,49-56,58,59] We’ve revised the introduction and discussion with the above information added.

Modified text, Pages 2-3, lines 101-103

‘… and (3) to perform a systematic review of previous reports about laryngeal H. pylori infection and laryngeal cancer, to determine the effect of laryngeal H. pylori infection on the development of laryngeal cancer.

-->

‘… and (3) to elucidate the impact of laryngeal H. pylori infection on the development of laryngeal cancer across countries by performing a systematic review on the topic.’

Modified text, Pages 12-13, lines 411-431

‘… Nevertheless, this systematic review still supported that laryngeal H. pylori infection, detected by PCR test, was associated with laryngeal cancer worldwide despite the low level of evidence. Therefore, we used three approaches to investigate the relationship between laryngeal H. pylori infection and laryngeal cancer in this hospital-based study.

-->

‘… Because of the heterogeneity in sample types, detection methods, and study designs of previous reports, it was difficult to conduct a multi-country meta-analysis to determine the effect of laryngeal H. pylori infection on the development of laryngeal cancer. We performed a systematic review in this study, which indicated that laryngeal H. pylori infection was associated with laryngeal cancer worldwide with a low level of evidence.

There were some interesting findings from the systematic review to be considered. First, the low prevalence (4.3%) of laryngeal H. pylori infection of the laryngeal malignancy, determined by a RUT, IHC, or histopathology, seemed not to support the role of H. pylori in the development of laryngeal cancer [12,21,49,50,54,59]. In contrast, laryngeal H. Pylori infection detected by PCR assay was associated with a “higher than expected” prevalence (40.0%) in patients with laryngeal cancer than that (25.9%) in controls [12,16,20,51-53,55-58]. Second, the FFPE samples of laryngeal cancer might be less suitable for detecting H. pylori infection than the fresh or fresh-frozen samples, because the obviously low detecting rate (2.6%) of the FFPE sample [12,16,21,57,59] comparing with that (49.7%) of fresh or fresh-frozen samples [20,49-56,58,59]. Notably, H. pylori infection in the laryngeal mucosa may play a role in the development of laryngeal cancer after destroying the mucosal lining of the larynx [20]. Therefore, to detect laryngeal H. pylori infection, examining fresh or fresh-frozen laryngeal samples with PCR assay may be the most optimal approach for patients with laryngeal lesions in order to stratify the risk of laryngeal cancer.’

  1. Figure 1 IHC data looks not highly significant. Authors should quantify the data and represent it as a bar diagram. Present the IHC image clearly by arrow showing what is the area where they want to show the difference. Also mention what is the control they used here.

REPLY. Thank you very much for these important comments. We’ve added arrows showing the presence of H. pylori antigen in the (a) positive control, (b) negative control, (c) H. pylori antigen-positive laryngeal polyp, and (d) H. pylori antigen-positive laryngeal cancer in these IHC images. For detecting H. pylori antigen, we considered qualitative evaluation rather than quantitative evaluation due to only two H. pylori antigen-positive patients. We appreciate your excellent suggestions and may present the difference in expression of H. pylori antigen in more cases in the future. We’ve revised Figure 1 with the information added.

Modified text, Page 7, lines 263-270

Figure 1. Representative examples of Helicobacter pylori (H. pylori) antigen detection using immunohistochemistry (IHC). (a) IHC staining of H. pylori antigen (brown) from a gastric biopsy in a patient with gastric adenocarcinoma. (b) IHC staining in the gastric tissue of a patient with H. pylori-negative gastric adenocarcinoma. (c) IHC staining of H. pylori antigen in the laryngeal tissue of a patient with vocal polyp. (d) IHC staining of H. pylori antigen in the laryngeal tissue of a patient with laryngeal squamous cell carcinoma. Original magnification: 40×.

-->

Figure 1. Representative examples of Helicobacter pylori (H. pylori) antigen detection using immunohistochemistry (IHC). (a) Positive IHC staining of H. pylori antigens (brown color) from a gastric biopsy in a patient with gastric adenocarcinoma and H. pylori infection (arrows). (b) Negative IHC staining in the gastric tissue of a patient with H. pylori-negative gastric adenocarcinoma. (c) Positive IHC staining of H. pylori antigen (brown color) in the laryngeal tissue of a patient with vocal polyp (arrow). (d) Positive IHC staining of H. pylori antigen (brown color) in the laryngeal tissue of a patient with laryngeal cancer (arrows). Original magnification: 40×.

  1. Add the PCR data as supplementary to support their conclusion. Provide the gel image as supplementary.

REPLY. Thank you very much for this in-depth suggestion. In this study, we have performed DNA sequencing for the presence of ureC, compared the nucleotides of DNA sequences by using the Basic Local Alignment Search Tool (BLAST®; http:// https://blast.ncbi.nlm.nih.gov/), and confirmed H. pylori infection (genotype: Puno135). We’ve revised the result with new Figure 2 added.

Modified text, Page 3, lines 146-148

‘… staining with ethidium bromide (0.5 mg/mL).’

-->

‘… staining with ethidium bromide (0.5 mg/mL). In the case of the presence of ureC, DNA sequencing was performed, and the nucleotides of DNA sequences were compared by using the Basic Local Alignment Search Tool (http:// https://blast.ncbi.nlm.nih.gov/).’

Modified text, Page 6, lines 255-258

‘Both the PCR and IHC approach detected a presence of H. pylori in the laryngeal lesions. The H. pylori DNA positive rates were low …’

-->

‘… able to determine the causal effect relationship between H. pylori and laryngeal cancer [70]. Lastly, this study focused on the presence of H. …’

Modified text, Page 7, lines 272-276

Add:

Figure 2. Helicobacter pylori DNA detection. (a) Gel electrophoresis of polymerase chain reaction product of ureC (136 bp) gene. (b) DNA sequencing for the presence of ureC in one patient with vocal polyp and laryngeal Helicobacter pylori infection (genotype: Puno135). Lane 2: positive ureC; Lane 1: negative ureC; P: positive control; N: negative control; M: 100-bp DNA marker.

Reviewer 4 Report

I read with interest the paper on the relationship between Hp and laryngeal neoplasias. I think the study is well designed, well conducted and well written. The methods are appropriate to answer the study hypotheses correctly provides in the results section. 

Results provided by the study are completed with a systematic review on the topic.

The main limit of the study is the Hp infection was tested only in the laryngeal tissue and presence of Hp in the stomach or at least UBT or Hp stool antigen was not performed. But authors are aware of this and appropriately discussed this point in the Discussion section.

Minor comments:

  • PRISMA guidelines should be included in the reference list
  • Figure 2 in the last rectangle it should be "quantitative" instead of "qualitative" 

Author Response

Reviewer 4’s comments:

I read with interest the paper on the relationship between Hp and laryngeal neoplasias. I think the study is well designed, well conducted and well written. The methods are appropriate to answer the study hypotheses correctly provides in the results section.

Results provided by the study are completed with a systematic review on the topic.

REPLY. We appreciate these encouraging comments. We amend the modified text concerning your suggestions. The quality of this revised manuscript has been substantially improved. Thank you very much!

The main limit of the study is the Hp infection was tested only in the laryngeal tissue and presence of Hp in the stomach or at least UBT or Hp stool antigen was not performed. But authors are aware of this and appropriately discussed this point in the Discussion section.

REPLY. Thank you very much for these in-depth comments. Because we did not test the presence of H. pylori in the stomach or at least UBT or H. pylori stool antigen in this study, the lack of H. pylori in laryngeal cancer could not be interpreted as a lack of contribution of H. pylori for the development of laryngeal cancer. Our findings mentioned the importance of PCR method and fresh/fresh-frozen samples for detecting laryngeal H. pylori infection. A further study will be warranted to investigate the association of gastric H. pylori infection and laryngeal cancer. We appreciate your cogent suggestions.

Minor comments:

  • PRISMA guidelines should be included in the reference list

REPLY. Thank you very much for this important suggestion. We’ve revised the method with the information added.

Modified text, Page 4, lines 179-182

To identify the association between H. pylori infection and laryngeal cancer in adults, we stated the clinical question as follows: can laryngeal H. pylori infection increase the risk of laryngeal cancer?’

-->

‘Systematic searches using the Preferred reporting items for systematic reviews guidelines (PRISMA) [47] to identify studies that reported the association between H. pylori infection and laryngeal cancer in adults were performed. Therefore, the clinical question as follows: ‘can laryngeal H. pylori infection increase the risk of laryngeal cancer?’ was stated.’

  • Figure 2 in the last rectangle it should be "quantitative" instead of "qualitative"

REPLY. We’ve revised Figure 2, thank you very much for pointing this out for us.

Modified text, Page 10, lines 357-358

Figure 2. The flow of study identification, inclusion, and exclusion.’

--> Amended Figure 2 as indicated.  

Reviewer 5 Report

This manuscript by Li-Jen Hsin et al. describes interesting data on the impact of Hp infection on Laryngeal cancers.

Nevertheless, this manuscript suffers from numerous imprecisions and a lack of data to be considered suitable for publication.

Global : 

Please italicize : "e.g", "et al."

Prefer passive form.

Number </= to 12 must be written in full letters.

Introduction : 

What does "old age" mean? (line 44)

Line 62 to 65 and global manuscript. It is important to distinguish viable Hp (obtained by culture) and nonviable/viable Hp obtained by molecular biology. Please complete

Line 81 to 83 is not necessary and complicates the understanding of the manuscript. please suppress.

Methods : 

How was determined the included number of patients ? This study seems to suffer from a lack of power. please complete.

How many readers (and what are their levels of expertise) are implicated in the histological score calculation (part 2.5)?

'Part 2.6.3 and 2.6.4How many articles were included in this third classification? Who was this 3rd reviewer?

Have the authors consider classifying the patients according to their Hp serology status for example? Have they received a gastric biopsy examination?

Results :

Table 2. Precise what represents N%.

Table 5. The table was cut. I cannot read the last column (and maybe the ones after this one.

Discussion : 

Line 323: Larynx or Pharynx?

Author Response

Reviewer 5’s comments:

This manuscript by Li-Jen Hsin et al. describes interesting data on the impact of Hp infection on Laryngeal cancers.

Nevertheless, this manuscript suffers from numerous imprecisions and a lack of data to be considered suitable for publication.

REPLY. We appreciate these encouraging and cogent comments. We amend the modified text concerning your suggestions. The quality of this revised manuscript has been substantially improved. Thank you very much!

Global :

Please italicize : "e.g", "et al."

REPLY. Thank you for this suggestion. We’ve amended in the revised manuscript.

Modified text, Page 5, line 199

‘… lows: (1) papers not written in English, (2) research not involving humans (e.g., in vitro or …’

-->

‘… lows: (1) papers not written in English, (2) research not involving humans (e.g., in vitro or …’

Prefer passive form.

REPLY. Thank you very much for this comment. We’ve revised the abstract, instruction, methods, and results accordingly.

Modified text, Page 1, lines 24-36

Helicobacter pylori (H. pylori) … However, whether H. pylori is a risk factor for laryngeal cancer is not conclusive. This study aimed to investigate the possible association of laryngeal H. pylori infection in patients with newly diagnosed laryngeal cancer and perform a systematic review of previous reports about H. pylori infection laryngeal cancer. … three subgroups. Data of patients with laryngeal cancer and controls from included 15 reports showed that the prevalence of laryngeal H. pylori infection were 29.4% and 16.7%, respectively. …’

-->

Helicobacter pylori (H. pylori) … However, laryngeal H. pylori infection in Taiwanese patients with newly diagnosed laryngeal cancer has not been reported. This study was aimed to investigate the possible association between laryngeal H. pylori infection and laryngeal cancer in Taiwan and perform a systematic review of previous reports in other countries. … three subgroups. Fifteen studies were involved in the systematic review of the digital literature database, distributed to February 2021. Data of patients with laryngeal cancer and controls showed that the laryngeal H. pylori infection rates were 29.4% and 16.7%, respectively. …’

Modified text, Page 2, line 96

‘… larynx - lesions) remains controversial [31]. The aims of this study were (1) …’

-->

‘… larynx - lesions) remains to be controversial [35]. The aims of this study were (1) …’

Modified text, Page 3, lines 106-110

We recruited 105 consecutive patients with newly diagnosed, … between August 1, 2012 and December 31, 2015. …’

-->

‘A total of 105 consecutive patients with newly diagnosed, … between 1st August 2012 and 31st December 2015 were recruited. …’

Modified text, Page 3, lines 118-119

‘… All participants provided written informed consent before enrolment. …’

-->

‘… Written informed consent were obtained from all the participants before enrolment. …’

Modified text, Page 3, lines 126-128

‘… Patients rated their severity of nine problems over the past month on a scale of 0 (no problem) to 5 (severe problem) with a maximum total score of 45. …’

-->

‘… The severities of nine problems over the past month on a scale of 0 (no problem) to 5 (severe problem) with a maximum total score of 45 were rated by patients. …’

Modified text, Page 3, lines 133-135

‘Using hematoxylin and eosin staining, we histologically evaluated the lesion type and tissue adequacy (≥10.0% mucosal lesion cells) and resected the corresponding formalin-fixed, paraffin-embedded (FFPE) tissues [32]. …’

-->

‘Using hematoxylin and eosin staining, the lesion type and tissue adequacy (≥10.0% mucosal lesion cells) and resected the corresponding formalin-fixed, paraffin-embedded (FFPE) tissues were histologically evaluated [36]. …’

Modified text, Page 4, lines 179-182

To identify the association between H. pylori infection and laryngeal cancer in adults, we stated the clinical question as follows: can laryngeal H. pylori infection increase the risk of laryngeal cancer?’

-->

‘Systematic searches using the Preferred reporting items for systematic reviews guidelines (PRISMA) [42] to identify studies that reported the association between H. pylori infection and laryngeal cancer in adults were performed. Therefore, the clinical question as follows: ‘can laryngeal H. pylori infection increase the risk of laryngeal cancer?’ was stated.’

Modified text, Page 4, lines 186-189

‘… We performed a detailed search using (“Helicobacter pylori” [Mesh] or “(H. pylori)” [Mesh] or “Helicobacter Infections” [Mesh] or “pylori” [Mesh]) and (“laryngeal cancer” [Mesh] or “laryngeal squamous cell carcinoma” [Mesh] or “larynx” [Mesh]). …’

-->

‘… A detailed search using (“Helicobacter pylori” [Mesh] or “(H. pylori)” [Mesh] or “Helicobacter Infections” [Mesh] or “pylori” [Mesh]) and (“laryngeal cancer” [Mesh] or “laryngeal squamous cell carcinoma” [Mesh] or “larynx” [Mesh]) was performed. …’

Modified text, Page 4, lines 193-195

Two reviewers (H.-H.C. and L.-A.L.) independently performed primary screening of the articles based on their titles and abstracts blinded to each other’s decisions. …’

-->

‘Primary screening of the articles based on their titles and abstracts was independently performed by two reviewers (H.-H.C. and L.-A.L.) who were blinded to each other’s decisions. …’

Modified text, Page 5, lines 211-212

Two reviewers (H.-H.C. and L.-A.L.) extracted data from each study separately using a data extraction form that was designed in advance. …’

-->

‘Data from each study were extracted by two reviewers (H.-H.C. and L.-A.L.) separately using a data extraction form that was designed in advance. …’

Modified text, Page 5, lines 234-235

We enrolled 93 (89%] men and 12 (11%) women with a mean age of 58.6 ± 14.0 years. …’

-->

‘Ninety-three (89%] men and twelve (11%) women with a mean age of 58.6 ± 14.0 years were enrolled. …’

Number </= to 12 must be written in full letters.

REPLY. Amended, thanks!

Modified text, Page 5, lines 213-214

We enrolled 93 (89%] men and 12 (11%) women with a mean age of 58.6 ± 14.0 years. …’

-->

‘Ninety-three (89%] men and twelve (11%) women with a mean age of 58.6 ± 14.0 years were enrolled. …’

Introduction :

What does "old age" mean? (line 44)

REPLY. Thank you for bringing this issue to us. The major risk factors for laryngeal cancer include cigarette smoking, alcohol consumption, male sex, and age > 55 years old. We’ve amended this sentence in the revised manuscript.

Modified text, Page 1, line 46

‘… male sex, and old age [4].’

-->

‘… male sex, and age > 55 years old [4].’

Line 62 to 65 and global manuscript. It is important to distinguish viable Hp (obtained by culture) and nonviable/viable Hp obtained by molecular biology. Please complete

REPLY. Thank you very much for this cogent comment. Currently, many techniques are being applied for the diagnosis of H. pylori infection. Bacterial culture is the most specific way to confirm viable H. pylori of the mucosa tissue; however, this method is tedious, time-consuming, and unnecessary for the routine diagnosis of H. pylori infection. Other invasive diagnostic methods (such as the rapid urease test, histological examination, and molecular examination) and non-invasive diagnostic methods (such as the urea breath test, stool antigen detection, or serology) have been widely used to diagnose H. pylori infection [DOI: 10.3748/wjg.v20.i28.9299]. However, these methods present non-optimum specificity (e.g., the rapid urease test, urea breath test, and serology) or difficulty in assessing the existence of active infection (e.g., histological examination, molecular examination, and stool antigen detection) [DOI: 10.3748/wjg.v21.i40.11221]. We’ve revised the introduction with the information added.

Modified text, Page 2, lines 68-83

‘Increasing evidence supports that H. pylori can play a role in the development of laryngeal cancer, including that (1) H. pylori can live in the normal larynx [10], vocal polyps (VP) [20], vocal fold leukoplakia (VFL) [21], and precancerous lesions [22], and (2) H. pylori can induce systemic inflammation and promote tumorigenesis [23].’

-->

‘Increasing evidence supports that H. pylori may play a role in the development of laryngeal cancer, including that (1) H. pylori can be detected in the normal larynx [16], vocal polyps (VP) [24], vocal fold leukoplakia (VFL) [25], and precancerous lesions [26], and (2) H. pylori can induce systemic inflammation and promote tumorigenesis [26]. A previous systematic review suggested that H. pylori infection of the stomach, larynx or non-specific sites was related to laryngeal cancer [17].

Noteworthily, H. pylori infection has been diagnosed by different techniques in those studies [16,17,24-26]. Bacterial culture is the most specific way to confirm viable H. pylori of the mucosa tissue; however, this method is tedious, time-consuming, and un-necessary for the routine diagnosis of H. pylori infection. Other invasive diagnostic methods (such as the rapid urease test [RUT], histological examination, immunohisto-chemistry [IHC], or polymerase chain reaction [PCR] assay) and non-invasive diagnostic methods (such as the urea breath test, stool antigen detection, or serology) have been widely used to diagnose H. pylori infection [27]. However, these methods present non-optimum specificity (e.g., the RUT, urea breath test, and serology) or difficulty in assessing the existence of active infection (e.g., histological examination, IHC, and PCR) [28].’

Line 81 to 83 is not necessary and complicates the understanding of the manuscript. please suppress.

REPLY. Thank you very much for this in-depth comment. We’ve suppressed this sentence in the revised manuscript.

Modified text, Page 2, lines 77-80

However, the role of H. pylori infection in the development of laryngeal cancer (inflamed larynx - lesions) remains controversial [30]. The aims of this study were (1) to investigate the prevalence of laryngeal H.

There were three hypotheses for this study: (1) laryngeal H. pylori infection is associated with laryngeal cancer, (2) expression of E-cadherin and CD1d of the larynx are related to laryngeal cancer, and (3) laryngeal H. pylori infection is a risk factor of laryngeal cancer.

-->

‘The role of gastric H. pylori infection in the pathogenesis of gastric cancer has been substantially studied. Some research has also suggested an association between laryngeal H. pylori infection and laryngeal cancer (inflamed larynx - lesions), however, the connection remains to be controversial [31]. The study hypothesis was that laryngeal H. pylori infection is associated with the development of laryngeal cancer. The aims of this study were (1) to investigate the prevalence of laryngeal H. …’

Methods :

How was determined the included number of patients ? This study seems to suffer from a lack of power. please complete.

REPLY. Thank you very much for bringing this issue to us. Sample sizes were estimated using a priori calculations [Fisher’s exact test, OR = 2.9, two-tailed α = 0.05, power = 0.90; allocation ratio = 0.4], as reported in a previous meta-analysis of H. pylori infection in patients with laryngeal cancer and controls (doi:10.1002/hed.24214). We needed to recruit a total of 105 patients for this study. We’ve revised the methods with the information added.

Modified text, Page 3, lines 110-113

‘… This study protocol has been reported elsewhere [32,33]. The inclusion criteria were as follows: age >18 years, a pathological diagnosis (VP, …’

-->

‘… This study protocol has been reported elsewhere [36,37]. The sample size (The = 105) was estimated using a priori calculations [Fisher’s exact test, OR = 2.9, two-tailed α = 0.05, power = 0.90; allocation ratio = 0.4], as reported in a previous meta-analysis of H. pylori infection in patients with laryngeal cancer and controls [17]. The inclusion criteria were as follows: age >18 years, a pathological diagnosis (VP, …’

How many readers (and what are their levels of expertise) are implicated in the histological score calculation (part 2.5)?

REPLY. Two scientific team members (T.-C.C. [professor of pathology; 25-year experience] and C.-G.C. [assistant professor of laboratory medicine; 25-year experience]) performed histological score calculations. We’ve revised the method with the information added.

Modified text, Page 4, lines 175-1176

‘… by the scientific team members without knowing the clinical in-formation [32,33].’

-->

‘… by the scientific team members (T.-C.C. and C.-G.H.) without knowing the clinical in-formation [36,37].’

'Part 2.6.3 and 2.6.4 How many articles were included in this third classification? Who was this 3rd reviewer?

REPLY. Thank you for these cogent comments. There was one article (doi:10.1002/hed.23492) that the two reviewers (H.-H.C. and L.-A.L.) could not decide to accept or reject the article due to case numbers of laryngeal lesions that were not directly reported. A third person (L.-J.H.) was invited to make a final decision in this third classification. We’ve revised the method with the information added.

Modified text, Page 5, lines 207-209

‘… A third person was invited to make a final decision when the two reviewers could not decide to accept or reject the article after discussion.’

-->

‘… A third person (L.-J.H.) was invited to make a final decision when the two reviewers could not decide to accept or reject the article after discussion (n = 1).’

Modified text, Page 5, lines 214-215

‘… When the two reviewers had unresolved opinions, a third person was invited to make a final decision.’

-->

‘… When the two reviewers had unresolved opinions, a third person (L.-J.H.) was invited to make a final decision (n = 1).’

Have the authors consider classifying the patients according to their Hp serology status for example? Have they received a gastric biopsy examination?

REPLY. Thank you very much for these cogent comments. In the original prospective study, serology tests of H. pylori and gastric biopsy examination were not included because the primary focus of this study was to detect H. pylori infection in laryngeal tissues. There were no patients with a serology test of H. pylori and 41 patients with a gastric biopsy examination after further medical chart reviews. However, these data were not sufficient to determine whether the patient was gastric H. pylori-naïve, currently H. pylori-infected, or with H. pylori past infection. We’ve revised the abstract with the information added.

Modified text, Page 13, lines 432-436

‘The primary focus of this study was to detect H. pylori infection in laryngeal tissues. The gastric H. pylori infection status was retrieved from medical chart reviews and evaluated as a reference, which was not sufficient to determine whether the patient was gastric H. pylori-naïve, currently H. pylori-infected, or with H. pylori past infection. Because …’

-->

‘The primary focus of this study was to detect H. pylori infection in laryngeal tissues. There were no patients with a serology test of H. pylori and 41 patients with a gastric biopsy examination after further medical chart reviews. Therefore, we could not determine whether the patient was gastric H. pylori-naïve, currently H. pylori-infected, or with H. pylori past infection. Because …’

Results :

Table 2. Precise what represents N%.

REPLY. Thank you for this cogent comment. We’ve revised Table 2 in the modified manuscript.

Modified text, Pages 7-8, Table 2

Variables

All

VP

VFL

Laryngeal cancer

p-Value 1

Helicobacter pylori status in the laryngeal lesions (n = 105)

N (%)

105 (100)

37 (35)

22 (21)

46 (44)

DNA positivity, n (%)

1 (1)

1 (3)

0 (0)

0 (0)

0.40

Antigen positivity, n (%)

2 (2)

1 (3)

0 (0)

1 (2)

0.75

Immunohistochemistry of Helicobacter pylori- and LPR-related biomarkers (n = 84)

N (%)

73 (100)

24 (33)

17 (23)

32 (44)

Expression of E-cadherin, mean (SD)

95.5 (71.1)

69.8 (55.8)

115.0 (65.2)

104.4 (80.3)

0.08

N (%)

84 (100)

26 (31)

17 (20)

41 (49)

Expression of CD1d, mean (SD)

41.8 (48.6)

23.0 (35.6)

37.7 (41.7)

53.0 (60.2)

0.18

 -->

Variables

All

VP

VFL

Laryngeal cancer

p-Value 1

Helicobacter pylori status in the laryngeal lesions (n = 105)

Patent number (%)

105 (100)

37 (35)

22 (21)

46 (44)

DNA positivity, n (%)

1 (1)

1 (3)

0 (0)

0 (0)

0.40

Antigen positivity, n (%)

2 (2)

1 (3)

0 (0)

1 (2)

0.75

Immunohistochemistry of Helicobacter pylori- and LPR-related biomarkers (n = 84)

Patent number (%)

73 (100)

24 (33)

17 (23)

32 (44)

Expression of E-cadherin, mean (SD)

95.5 (71.1)

69.8 (55.8)

115.0 (65.2)

104.4 (80.3)

0.08

Patent number (%)

84 (100)

26 (31)

17 (20)

41 (49)

Expression of CD1d, mean (SD)

41.8 (48.6)

23.0 (35.6)

37.7 (41.7)

53.0 (60.2)

0.18

Table 5. The table was cut. I cannot read the last column (and maybe the ones after this one.

REPLY. Thank you very much for bringing the issue to us. We rechecked the original manuscript (word format) and found Table 5 was not cut. Perhaps this issue was caused by PDF transformation. Therefore, we have shortened the lengths of the columns to reduce the table size for clear reading.

Modified text, Page 11, Table 5 (Please read the modified text, thanks!)

Discussion :

Line 323: Larynx or Pharynx?

REPLY. Thank you very much for bringing the issue to us. We’ve amended this sentence in the revised manuscript.

Modified text, Page 12, line 367

‘… carcinoma of the larynx of pharynx showed that H. pylori infection, diagnosed by PCR or …’

-->

‘… carcinoma of the larynx and pharynx showed that H. pylori infection, diagnosed by PCR or …’

Round 2

Reviewer 3 Report

The authors rectify all of my queries and incorporate the information in the updated manuscript. All their replies are satisfactory. The manuscript is nore ready for publication.

Reviewer 5 Report

After this extensive revision, the manuscript is now suitable for publication.